

# Clinical characteristics and prognostic nomogram analysis of patients with dual primary cancers with first gastric cancer: a retrospective study in China

Bing Wang[1] and Lu Liu[2,3]

[1] The First Clinical Medical College of Gansu University of Chinese Medicine, Lanzhou University, Lanzhou, Gansu, China
[2] The First School of Clinical Medicine, Lanzhou University, Lanzhou, Gansu, China
[3] Department of Otolaryngology, Head and Neck Surgery, Gansu Provincial Hospital, Lanzhou, Gansu, China

Corresponding author
Lu Liu, liulu9630@163.com

## ABSTRACT

**Background**. With the improvement in diagnosis and treatment of gastric cancer (GC), the survival time of patients has been gradually prolonged. However, these survivors are at increased risk for other diseases, including second primary cancers (SPCs). Currently, there remain few central studies concerning double primary cancers with first gastric cancer (DPCFGC). Thus, this study aimed to investigate these patients' clinical characteristics and perform prognostic nomogram analysis.

**Methods**. The clinical data of 78 DPCFGC patients were retrospectively collected and analyzed through the hospital electronic medical record system. Univariate and multivariate Cox regression analyses were performed to screen independent risk factors, based on which the prognostic nomogram was further constructed and validated using the R software package. Finally, Kaplan–Meier curves were plotted to explore the association of overall survival (OS) with prognostic factors and the model.

**Results**. The prevalence of DPCFGC was 0.86%, of which the proportions of synchronous and metachronous patients were 47.44% and 52.56%, respectively; 65.38% (51/78) and 34.62% (27/78) of patients were male and female, respectively. The median age at GC and SPC diagnosis was 63 and 65 years, respectively, and 52.57% of GC patients developed SPCs within 1 year. The top three SPCs were in the esophagus (19.24%), colon (16.67%), and rectum (15.39%). The most common features of GC and SPCs were adenocarcinoma, poorly and moderately differentiated histology, and pathological stages I and II. The radical resection rate of GC was significantly lower in synchronous patients than in metachronous patients (45.94% *vs.* 100.00%, $P < 0.001$), but no significant difference was noted in the radical resection rate of SPCs (35.13% *vs.* 46.34%, $P = 0.315$). The OS of DPCFGC patients was $31.03 \pm 4.14$ months. The pathological stage of GC and SPCs, whether to operate for GC, and diagnostic interval were independent risk factors. The predictive efficacy of the prognostic nomogram for 1-, 2- and 3-year OS in DPCFGC patients was 0.922, 0.935 and 0.796, respectively, with good consistency and clinical applicability. The OS was significantly lower in the high-risk group than in the low-risk group.

**Conclusions**. During follow-up, clinicians should attach great importance to the screening of GC survivors, especially at early stage in older men within 1 year after diagnosis, and be alert to the possibility of occurrent digestive system malignancies.

The nomogram constructed in this study can provide a theoretical basis for the early clinical development of individualized treatment plans.

## INTRODUCTION

With technological advances in early diagnosis and treatment, the long-term survival rate of gastric cancer (GC) patients has improved, and the number of cancer survivors is increasing each year (*Sung et al., 2021*). However, these survivors are at a high risk of developing other diseases, including multiple primary malignant neoplasms (MPMNs) (*Miller et al., 2019*), defined as two or more unrelated primary cancers that develop simultaneously or sequentially in a patient, with double primary cancers being the most common (*Vogt et al., 2017*). In a European cancer survey, *Rosso et al. (2009)* reported that stomach was the fifth most common first primary cancer (FPC) site among MPMNs. In a 27-year follow-up study of GC patients in the United States, *Morais et al. (2017)* reported that 4.0% of patients had occurrent second primary cancers (SPCs). Clinicals should therefore to pay more attention to patients with potential MPMNs among GC survivors.

However, because MPMNs with first GC have a sporadic clinical onset, published studies are mostly case reports; moreover, there is a lack of standardized international guidelines for the diagnosis and treatment of MPMNs with first GC; hence, clinicians have an insufficient understanding of the disease (*Miller et al., 2019*; *Vogt et al., 2017*). Additionally, some studies that were based on American public databases could not accurately predict the prognosis of MPMNs patients owing to the exclusion of important information such as patients' lifestyle and SPCs' characteristics (*Chen, Sun & Liu, 2022*; *Wen et al., 2021*; *Bian et al., 2021*). Therefore, in order to guide the clinical diagnosis and treatment of such patients, it is necessary to analyze their clinical characteristics and establish a comprehensive prognostic model. Nomogram may be a good choice, which can integrate multiple independent predictors to accurately predict clinical outcomes, and has been widely used in the medical field (*Bianco Jr, 2006*; *Guillonneau, 2007*).

In this study, we selected the most common patients with dual primary cancer as subjects. We comprehensively screened, sorted, and analyzed the clinical characteristics of patients with double primary cancers with first GC (DPCFGC). Using the Cox proportional hazards model, we screened independent risk factors, based on which we further established and validated a prognostic nomogram to accurately predict the 1-, 2-, and 3-year overall survival (OS) of DPCFGC patients. Finally, we performed Kaplan–Meier analysis to explore the association of OS with prognostic factors and the model. Our study aimed to improve clinicians' awareness of high-risk DPCFGC patients among GC patients and provide a basis for formulating risk-matched clinical treatment planning.
## MATERIALS & METHODS

### Definition of MPMNs

The following definition criteria for MPMNs formulated by the International Association of Cancer Registries/International Institute of Cancer Research was adopted (*Curado et al., 2005*): (1) each tumor was confirmed as malignant by pathological biopsy; (2) each tumor was independent and different in anatomy; (3) MPMNs can occur regardless of the pathological morphology and time interval of tumors but cannot occur as a result of recurrences or metastases of FPC; and (4) discontinuous cancer foci with the same pathological type in the same organ was considered as multifocal cancers rather than MPMNs. When the diagnosis interval between two cancers was ≤ 6 months, the tumors were named as synchronous MPMNs, and when the interval was > 6 months, the tumors were named as metachronous MPMNs.

### Study design

Patients who met the definition criteria for MPMNs and were hospitalized in Gansu Provincial Hospital between January 2010 and January 2020 were screened in the hospital electronic medical records system. Five patients with missing relevant data were excluded, among which one case could not be diagnosed as SPCs only by imaging without biopsy, two cases had unknown stage because SPCs were hematological malignances, and two cases had unknown stage and treatment because GC was diagnosed in other hospitals. The selection procedure is illustrated as a flow diagram in Fig. 1. In total, 78 DPCFGC patients were identified on screening. The study was approved by Gansu Provincial Hospital (No. 2021-221). The need for patient consent was waived by the ethics committee owing to retrospective nature of the study and full data anonymization.

### Data collection and follow-up

The demographic data of patients and clinical information of two tumors were collected, such as gender, smoking history, alcohol consumption history, family history of cancers, history of chronic diseases (*e.g.*, coronary heart disease, diabetes and chronic obstructive pulmonary disease (COPD)), and characteristics of GC and SPCs. The DPCFGC patients were followed up though telephone inquiry until June 2022. With GC diagnosis as the starting point and patient death or follow-up cut-off as the end point, OS, median survival time (MST), and 1-, 2-, and 3-year survival rates were calculated.

### Statistical analysis

Data were analyzed and plotted by SPSS 26.0 software (version 26.0; SPSS, Chicago, IL, USA), GraphPad software (version 8.0; GraphPad, San Diego, CA, USA), and R software (version 4.1.3; *R Core Team, 2022*). The $\chi^2$ test or Fisher exact probability test was used to compare difference between categorical variables. Univariate and multivariate Cox regression analyses were used to determine independent risk factors, and the corresponding hazard ratio (HR) and 95% confidence intervals (CIs) were calculated. Risk scores of DPCFGC patients were calculated using the following formula: risk scores $= \beta1 \times 1 + \beta2 \times 2 + \cdots + \beta nXn$ (where $\beta$ is the prognostic regression coefficient and X is the predictor).

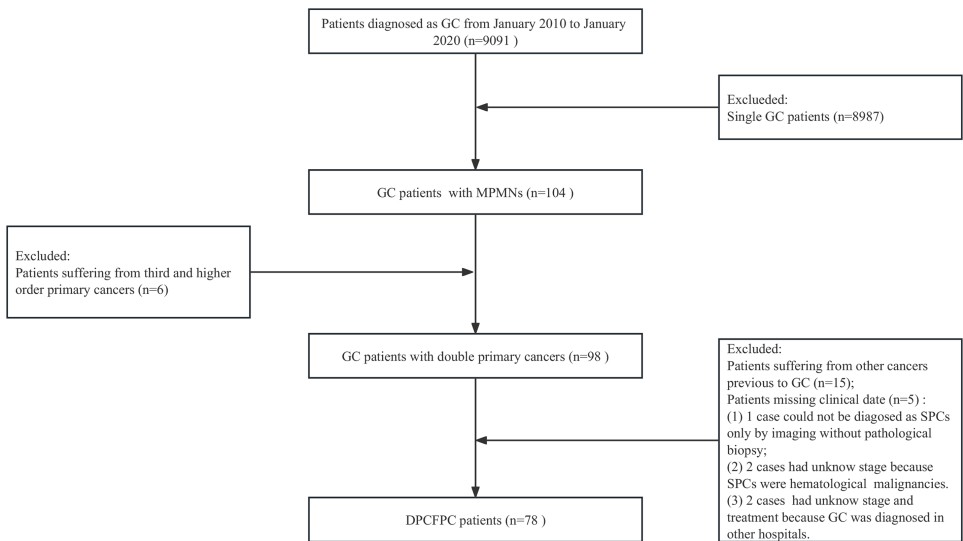

**Figure 1** **The flow chart of the screening process for DPCFPC patients.** DPCFPC, double primary cancer with first gastric cancer; GC, gastric cancer.

The predictors in our study included all prognostic factors that were significant in the multivariate Cox regression analysis, namely, pathological stage of GC ($P = 0.017$), whether to operate for GC ($P = 0.011$), diagnostic interval ($P = 0.006$) and pathological stage of SPCs ($P = 0.034$). Their prognostic regression coefficients were 0.557, −1.734, −0.600 and 0.514, respectively. Subsequently, we constructed a predictive prognostic nomogram. Receiver operating characteristic (ROC) curves, calibration curves and decision curve analysis (DCA) were plotted respectively to assess predictive performance, degree of fit and clinical benefit. Using the X-tile method, the optimal cutoff value for risk scores was determined, and based on this value, patients were divided into high-risk and low-risk groups. Kaplan–Meier curves were plotted to analyze the association of OS with prognostic factors and the model, and log-rank test was used to compare survival difference. R software packages used in this study included survival, survminer, rms, timeROC, and stdCa. A statistically significant difference was indicated at $P < 0.05$.

# RESULTS

## General clinical features of DPCFGC patients

Overall, 9091 GC patients were hospitalized in Gansu Provincial Hospital between January 2010 and January 2020. Among them, 78 patients had DPCFGC, with the prevalence rate being 0.86%. Among them, 52.56% (41/78) were metachronous patients and 47.44% (37/78) were synchronous patients; 65.38% (51/78) were male and 34.62% (27/78) were female. The median age at GC and SPCs diagnosis was 63 and 65 years, respectively (Fig. 2A). Analysis of the diagnostic interval showed that the incidence rates for SPCs at 6 months, 1 year, 3 years, and 5 years after GC were 47.44%, 52.57%, 83.34%, and 97.44%, respectively (Fig. 2B), suggesting that more than half of the SPC cases occurred within the

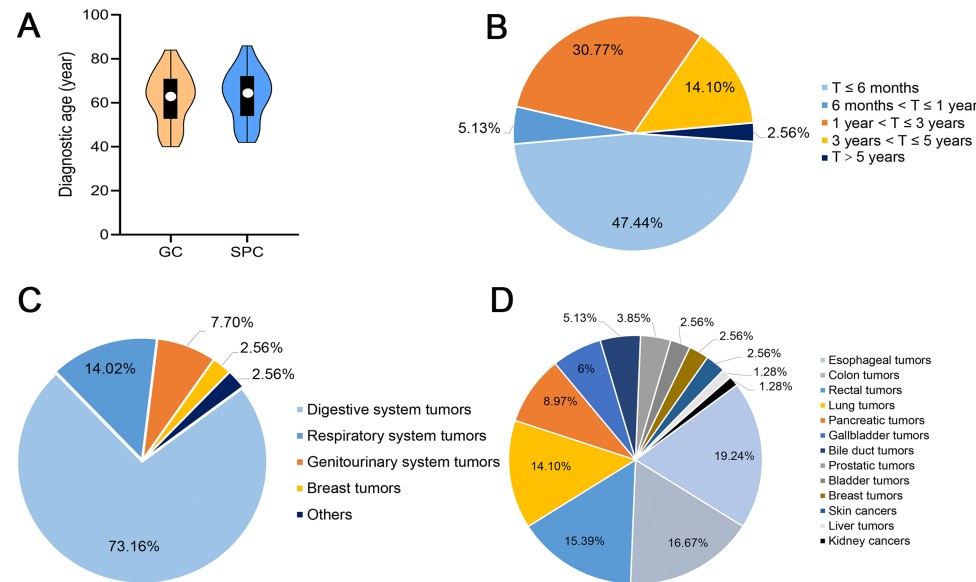

**Figure 2  General clinical features of DPCFGC patients.** (A) Age at diagnosis of GC and SPCs. (B) Diagnostic interval between GC and SPCs. (C) Systems involved in SPCs. (D) Organs involved in SPCs. DPCFGC, double primary cancer with first gastric cancer; GC, gastric cancer; SPCs, second primary cancers.

first year after diagnosis. Five systems were involved in SPCs, with the most common one being the digestive system (73.16%; Fig. 2C). Moreover, 13 organs were involved, with the top three being the esophagus (19.24%), colon (16.67%), and rectum (15.39%; Fig. 2D). As for the general characteristics, 24.36% (19/78) of patients had a smoking history, 20.51% (16/78) had an alcohol consumption history, 6.41% (5/78) had a family history of cancer, and 33.33% (26/78) had a history of chronic diseases.

## Pathological characteristics of GC and SPCs

The pathological characteristics of two tumors in DPCFGC patients were assessed, and the data are listed in Table 1. The most common characteristic features in both GC and SPCs were adenocarcinoma (91.03% *vs.* 70.51%), poorly and moderately differentiated histology (94.87% *vs.* 75.64%), and pathological stages I and II (56.41% *vs.* 52.56%). Subgroup analysis revealed no significant difference between synchronous and metachronous patients in the pathological type of GC ($P = 0.084$), histological grade of GC ($P = 0.683$), pathological type of SPCs ($P = 0.588$), and histological grade of SPCs ($P = 0.995$). Significant differences were noted only for the pathological stage of GC ($P < 0.001$) and pathological stage of SPCs ($P = 0.003$). Compared with synchronous patients, metachronous patients had an earlier pathologic stage of GC and SPCs.

## Treatment modalities for GC and SPCs

GC treatment involved surgery, chemotherapy, surgery + chemotherapy, and nutritional support. The radical resection rate of GC was 74.36% (58/78), adjuvant chemotherapy rate was 32.05% (25/78), and nutritional support rate was 17.95% (14/78). SPC treatment

**Table 1** Pathological characteristics of GC and SPCs in DPCFGC patients.

| Characteristics | DPCFGC | Synchronous patients | Metachronous patients | *P*-value |
|---|---|---|---|---|
| Pathological type of GC | | | | 0.084 |
|    Adenocarcinoma | 71 (91.03%) | 31 (83.78%) | 40 (97.56%) | |
|    Others | 7 (8.97%) | 6 (16.22%) | 1 (2.44%) | |
| Histological grade of GC | | | | 0.683 |
|    Poor + moderate | 74 (94.87%) | 35 (94.59%) | 39 (95.12%) | |
|    High | 4 (5.13%) | 2 (5.41%) | 2 (4.88%) | |
| Pathological stage of GC | | | | <0.001[*] |
|    I–II | 44 (56.41%) | 12 (32.43%) | 32 (78.05%) | |
|    III | 34 (43.59%) | 25 (67.57%) | 9 (21.95%) | |
| Pathological type of SPCs | | | | 0.588 |
|    Adenocarcinoma | 55 (70.51%) | 25 (67.57%) | 30 (73.17%) | |
|    others | 23 (29.49%) | 12 (32.43%) | 11 (26.83%) | |
| Histological grade of SPCs | | | | 0.995 |
|    Poor + moderate | 59 (75.64%) | 28 (75.68%) | 31 (75.61%) | |
|    High | 19 (24.36%) | 9 (24.32%) | 10 (24.39%) | |
| Pathological stage of SPCs | | | | 0.003[*] |
|    I–II | 41 (52.56%) | 13 (35.14%) | 28 (68.29%) | |
|    III–IV | 37 (47.44%) | 24 (64.86%) | 13 (31.71%) | |

Notes.
DPCFGC, double primary cancer with first gastric cancer; GC, gastric cancer; SPCs, second primary cancers.
*$P<0.05$.

involved surgery, chemotherapy, radiotherapy, surgery + chemotherapy, surgery + radiotherapy, and nutritional support. The radical resection rate of SPCs was 41.03% (32/78), adjuvant chemotherapy rate was 37.18% (29/78), adjuvant radiotherapy rate was 7.69% (6/78), and nutritional support rate was 39.02% (16/78). Subgroup analysis revealed that the radical resection rate of GC in synchronous patients was significantly lower than that in metachronous patients (45.94% *vs.* 100.00%, $P < 0.001$), and no significant difference was observed in the radical resection rate of SPCs between two groups (35.13% *vs.* 46.34%, $P = 0.315$; Table 2).

## Analyses of survival status and prognostic factors

Using the Kaplan–Meier method, we analyzed the survival status of DPCFGC patients. The OS and MST of DPCFGC patients were 31.03 ±4.14 months and 26.00 months, respectively, with 1-, 2-, and 3-year survival rates of 61.54%, 48.72%, and 28.21%. In subgroup analysis, the OS and MST of synchronous patients were 13.00 ±4.00 months and 6.00 months, respectively, with a 3-year survival rate of 10.81%. The OS and MST of metachronous patients were 53.00 ±10.45 months and 35.00 months, respectively, with a 3-year survival rate of 43.90%. The survival status of metachronous patients was significantly better than that of synchronous patients ($P < 0.001$; Table 3).

Using Cox proportional hazards regression, we explored risk factors affecting the survival time of DPCFGC patients (Table 4). Univariate analysis revealed that smoking history ($P = 0.033$, HR = 1.813), alcohol consumption history ($P = 0.031$, HR = 1.883),

**Table 2  Treatment modalities of GC and SPCs in DPCFGC patients.**

| Therapy methods | DPCFGC | Synchronous patients | Metachronous patients |
|---|---|---|---|
| GC | | | |
|    Surgery | 39 (50.00%) | 9 (24.32%) | 30 (73.17%) |
|    Chemotherapy | 6 (7.69%) | 6 (16.22%) | 0 (0.00%) |
|    Surgery + chemotherapy | 19 (24.36%) | 8 (21.62%) | 11 (26.83%) |
|    Nutritional support | 14 (17.95%) | 14 (37.84%) | 0 (0.00%) |
| SPCs | | | |
|    Surgery | 11 (14.10%) | 5 (13.51%) | 6 (14.63%) |
|    Chemotherapy | 10 (12.82%) | 7 (18.92%) | 3 (7.32%) |
|    Radiotherapy | 4 (5.13%) | 1 (2.70%) | 3 (7.32%) |
|    Surgery + chemotherapy | 19 (24.36%) | 8 (21.62%) | 11 (26.83%) |
|    Surgery + radiotherapy | 2 (2.56%) | 0 (0.00%) | 2 (4.88%) |
|    Nutritional support | 32 (41.03%) | 16 (43.25%) | 16 (39.02%) |

Notes.
DPCFGC, double primary cancer with first gastric cancer; GC, gastric cancer; SPCs, second primary cancers.

**Table 3  Survival status of DPCFGC patients.**

| Variables | DPCFGC | Synchronous patients | Metachronous patients |
|---|---|---|---|
| OS (months) | 31.03 ± 4.14 | 13.00 ± 4.00 | 53.00 ± 10.45 |
| MST (months) | 26 | 6 | 35 |
| 1-year survival rate | 61.54% | 40.54% | 90.24% |
| 2-year survival rate | 48.72% | 21.62% | 82.93% |
| 3-year survival rate | 28.21% | 10.81% | 43.90% |

Notes.
DPCFGC, double primary cancer with first gastric cancer; OS, overall survival; MST, median survival time.

history of coronary heart disease ($P = 0.006$, HR = 3.174), history of COPD ($P = 0.027$, HR =3.960), age at GC diagnosis ($P = 0.027$, HR = 1.726), pathological stage of GC ($P = 0.004$, HR =2.089), whether to operate for GC ($P < 0.001$, HR = 0.098), diagnostic interval ($P < 0.001$, HR = 0.301), pathological stage of SPCs ($P = 0.001$, HR = 2.209), and whether to operate for SPCs ($P = 0.040$, HR = 0.605) were the significant risk factors that influenced the survival time of DPCFGC patients. We further performed a multivariate analysis of these indicators. Pathological stage of GC ($P = 0.017$, HR = 2.063), whether to operate for GC ($P = 0.011$, HR = 0.283), diagnostic interval ($P = 0.006$, HR = 0.359), and pathological stage of SPCs ($P = 0.034$, HR = 1.853) were significant independent risk factors for the prognosis of DPCFGC patients.

## Construction and validation of the prognostic Nomogram

Based on the results of regression analysis, we developed a prognostic nomogram (Fig. 3) and plotted ROC curves (Figs. 4A–4C), calibration curves (Figs. 4D–4F), and DCA (Figs. 4G–4I) for internal validation. Time-dependent ROC curves revealed that the area under the curve for predicting 1-, 2-, and 3-year OS was 0.922 (95% CI [0.864–0.979]), 0.935 (95%

**Table 4   Univariate and multivariate Cox analysis of factors associated with OS.**

| Variables | Univariate analysis | | Multivariate analysis | |
|---|---|---|---|---|
| | HR (95%CI) | *P*-value | HR (95%CI) | *P*-value |
| Gender (female *vs.* male) | 0.772 (0.469–1.269) | 0.307 | | |
| BMI ( >25 *vs.* ≤25) | 0.850 (0.342–2.115) | 0.727 | | |
| Smoking (yes *vs.* no) | 1.813 (1.049–3.135) | 0.033* | 2.115 (0.953–4.695) | 0.066 |
| Alcohol consumption (yes *vs.* no) | 1.883 (1.060–3.343) | 0.031* | 2.198 (0.972–4.972) | 0.059 |
| Family history of cancers (yes *vs.* no) | 0.990 (0.397–2.468) | 0.983 | | |
| History of hypertension (yes *vs.* no) | 1.851 (0.980–3.497) | 0.058 | | |
| History of coronary heart disease (yes *vs.* no) | 3.174 (1.403–7.180) | 0.006* | 1.054 (0.337–3.296) | 0.928 |
| History of diabetes (yes *vs.* no) | 1.908 (0.684–5.319) | 0.217 | | |
| History of COPD (yes *vs.* no) | 3.960 (1.171–13.384) | 0.027* | 1.056 (0.201–5.552) | 0.948 |
| Age at diagnosis of GC ( >60 *vs.* ≤60) | 1.726 (1.066–2.796) | 0.027* | 1.746 (0.995–3.064) | 0.052 |
| Histological grade of GC (high *vs.* poor + moderate) | 0.667 (0.209–2.1228) | 0.494 | | |
| Pathological stage of GC (III *vs.* I + II) | 2.089 (1.274–3.424) | 0.004* | 2.063 (1.139–3.736) | 0.017* |
| Whether to operate for GC (yes *vs.* no) | 0.098 (0.048–0.202) | <0.001* | 0.283 (0.108–0.747) | 0.011* |
| Whether adjuvant therapy for GC (yes *vs.* no) | 0.925 (0.559–1.532) | 0.763 | | |
| Diagnostic interval ( >6 *vs.* ≤6) | 0.301 (0.184–0.490) | <0.001* | 0.359 (0.173–0.745) | 0.006* |
| Age at diagnosis of SPCs ( >65 *vs.* ≤65) | 1.414 (0.887–2.252) | 0.145 | | |
| Histological grade of SPCs (high *vs.* poor + moderate) | 1.514 (0.892–2.570) | 0.125 | | |
| Pathological stage of SPCs (III + IV *vs.* I + II) | 2.209 (1.369–3.564) | 0.001* | 1.853 (1.048–3.279) | 0.034* |
| Whether to operate for SPCs (yes *vs.* no) | 0.605 (0.374–0.977) | 0.040* | 0.690 (0.376–1.267) | 0.232 |
| Whether adjuvant therapy for SPCs (yes *vs.* no) | 0.795 (0.497–1.270) | 0.336 | | |

**Notes.**

OS, overall survival; BMI, body mass index; COPD, chronic obstructive pulmonary disease; GC, gastric cancer; SPCs, second primary cancers.

*$P < 0.05$.

CI [0.877–0.994]), and 0.796 (95% CI [0.704–0.887]), respectively, which suggested that the prognostic model was more effective in predicting early OS in DPCFGC patients (Figs. 4A–4C). Calibration curves indicated that the prognostic model had good consistency with the actual results (Figs. 4D–4F). DCA results showed that the model had good clinical application value (Figs. 4G–4I).

## Association of OS with prognostic factors and the model

We explored the association between different prognostic predictors and the OS of DPCFGC patients by Kaplan–Meier analysis. The more advanced the pathological stage of GC (Fig. 5A), the non-radical surgery for GC (Fig. 5B), the diagnostic interval being

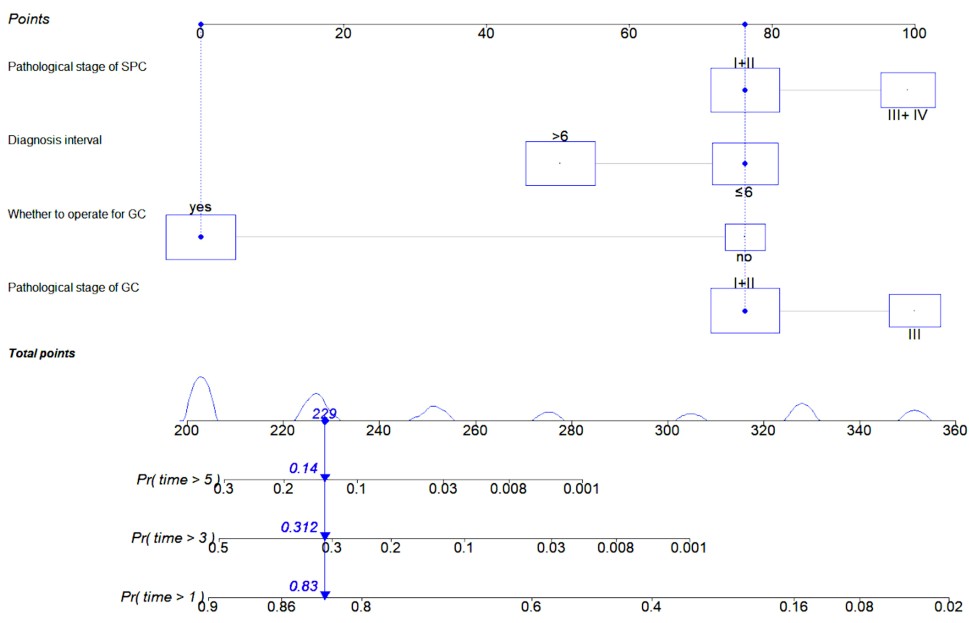

**Figure 3 Prognostic nomogram for predicting 1-, 2-, 3-year OS of DPCFGC patients.** DPCFGC, double primary cancer with first gastric cancer; GC, gastric cancer; SPCs, second primary cancers; OS, overall survival; Pr, probability.

less than 6 months (Fig. 5C), and the more advanced the pathological stage of SPCs (Fig. 5D), the shorter the OS was in DPCFGC patients. Using the X-tile method, we obtained the optimal cutoff value (0.168) for risk scores of the prognostic nomogram, and based on this value, we divided DPCFGC patients into low-risk and high-risk groups. Kaplan–Meier curves suggested that the OS was significantly lower in patients of the high-risk group than in patients of the low-risk group (Fig. 5F).

## DISCUSSION

With technological advances in the diagnosis and treatment of GC, the number of survivors with new primary malignant tumors has been increasing each year (*Sung et al., 2021*). According to the latest cancer report, the incidence of GC together with MPMNs ranges from 3.4% to 7.3% (*Zheng et al., 2021*), and the 10-year cumulative mortality due to GC with MPMNs is 1.5–2 times higher than that due to single GC (*Morais et al., 2020a*). However, because of the sporadic clinical onset of GC with MPMNs and the lack of international standardized treatment guidelines, clinicians have a poor understanding of this disease. Moreover, some studies based on the Surveillance, Epidemiology, and End Results database could not accurately predict disease prognosis because they did not include important information about patients' lifestyle (*e.g.*, history of smoking and alcohol consumption) and SPCs' characteristics (*e.g.*, pathological features and treatment status); consequently, clinicians are unable to formulate an optimal treatment plan for the affected individuals (*Bian et al., 2021*; *Wen et al., 2021*; *Chen, Sun & Liu, 2022*). In the present study, we comprehensively analyzed the clinical characteristics of DPCFGC patients

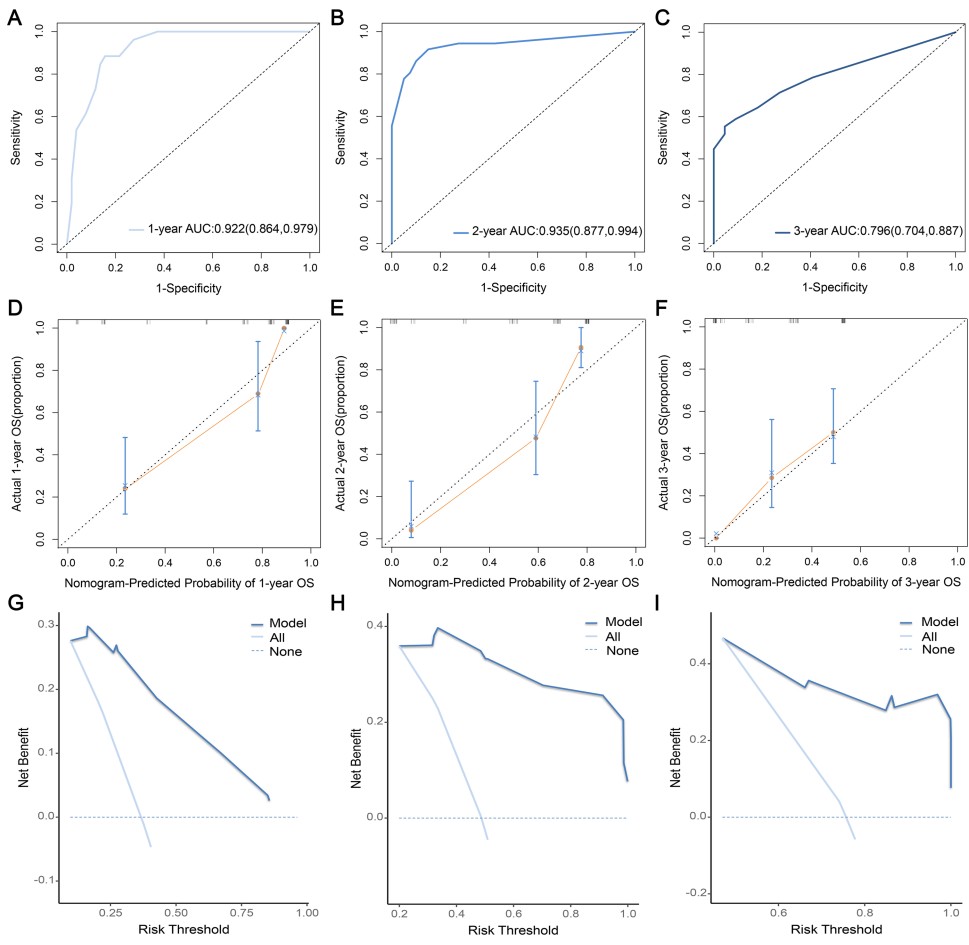

**Figure 4** **Evaluation of prognostic nomogram.** ROC curves for assessing the predictive efficacy at 1- (A), 2- (B), and 3-year (C). Calibration curves for assessing the degree of fit to actual results at 1- (D), 2- (E), 3-year (F). DCA for assessing the clinical benefit of operating characteristic; AUC, aera under the curves; DCA, the nomogram at 1- (G), 2- (H), 3-year (I). ROC, receiver decision curve analysis.

and constructed a nomogram model based on the screened independent risk factors. To the best of our knowledge, this is the first nomogram by integrating clinical data to analyze the prognosis of DPCFGC patients in a detailed manner and to perform subgroup analysis (synchronous and metachronous) based on the diagnostic interval, in order to improve clinicians' awareness and understanding of high-risk DPCFGC patients among GC patients in Asia and provide a basis for formulating risk-matched clinical treatment plans.

First, we analyzed the two tumors in terms of gender, age at diagnosis, diagnostic interval, sites of involvement, pathological features, and treatment modalities. Among all DPCFGC patients, 65.38% (51/78) were male, and 34.62% (27/78) were female. The median age at GC and SPCs diagnosis was 63 years and 65 years, respectively. Analysis of the diagnostic interval showed that more than half of the SPC cases occurred within the first year after diagnosis. These results are similar to those of previous studies. *Kim et al. (2016)* reported that patient age >63 years was an independent risk factor for GC with
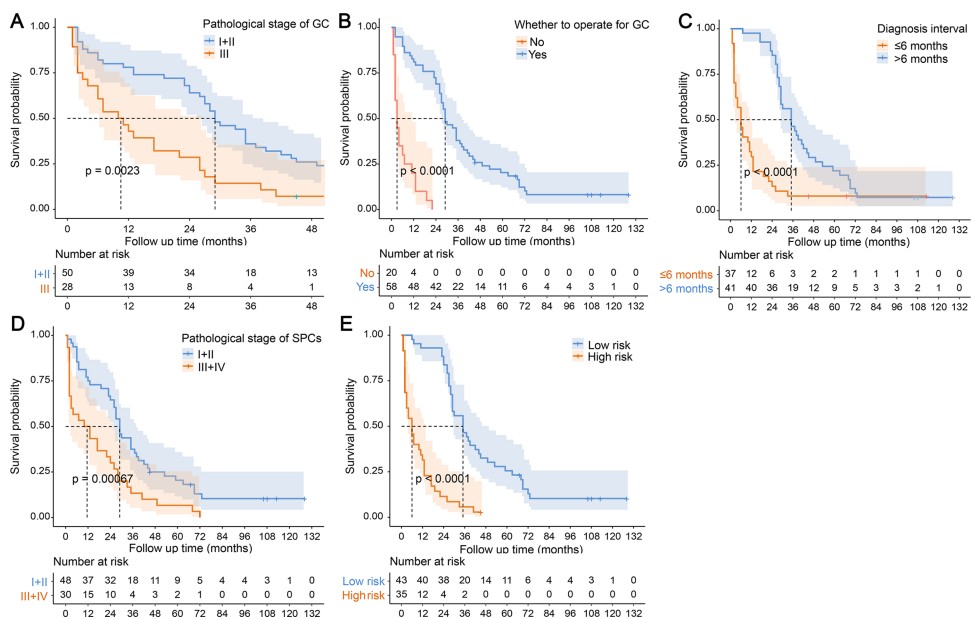

**Figure 5** **Association of OS with prognostic factors and the model by Kaplan–Meier analysis.** (A) Kaplan–Meier curves of patients with different pathological stage of GC. (B) Kaplan–Meier curves of patients with whether to operate for GC. (C) Kaplan–Meier curves of patients with different diagnostic interval. (D) Kaplan–Meier curves of patients with different pathological stage of SPCs. (E) Kaplan–Meier curves of patients with different risk. DPCFGC, double primary cancer with first gastric cancer; GC, gastric cancer; SPCs, second primary cancers; OS, overall survival.

occurrent colon cancer. *Morais et al. (2020b)* reported that the risk of occurrent GC within 2 months of FPC diagnosis was high, especially in survivors of esophageal, colon, and rectal cancers, with an incidence rate as high as 21.2%. We performed an in-depth literature search and found that the male population may be more vulnerable to the occurrence of SPCs than the female population. A study conducted in northern Portugal showed that the 10-year cumulative incidence rate for occurrent metachronous SPCs in GC patients was 5.7% in men and 3.5% in women. The standardized incidence rates for any type of SPCs in male and female GC patients were 1.30 (95% CI [1.11–1.52]) and 1.20 (95% CI [0.94–1.51]), respectively (*Morais et al., 2017*). A Korean study showed that male gender was an independent risk factor for occurrent colorectal cancer in GC patients (odds ratio [OR]: 2.933, 95% CI [1.307–6.584]) (*Kim et al., 2016*). A Japanese study showed that male gender was a risk factor for occurrent synchronous esophageal cancer in GC patients (OR: 13.3, 95% CI [1.8–96.6]) (*Ito et al., 2021*). These results suggest that clinicians should pay more attention to the follow-up of GC survivors in future work, especially elderly men, within 1 year after diagnosis.

Our study also showed that the digestive system was the most common system involved in SPCs, and the top three organs involved were esophagus, colon, and rectum, consistent with the findings of previous studies from Portugal and Poland (*Lawniczak et al., 2014*; *Morais et al., 2017*). We believe that this finding may be attributed to the high incidence of gastrointestinal tumors, shared risk factors, and clinical over-examination. Some of the

pathogenetic processes and risk factors are common in esophageal cancer, GC, colon cancer, and rectal cancer, including smoked food, alcohol consumption, and poor dietary habits (*Mysuru Shivanna & Urooj, 2016*). In addition, survivors of gastrointestinal tumors would have been required or self-required to undergo regular endoscopy to prevent recurrence, which undoubtedly increased the detection rate of digestive system malignancies in the cohort (*Pacheco-Figueiredo & Lunet, 2014*). Therefore, we should pay more attention to the possibility of occurrence of other cancers, especially digestive system tumors, among GC survivors. Such patients should be followed up and screened regularly.

Analysis of tumor pathological characteristics revealed a significant difference between synchronous and metachronous patients only in the pathological stage of GC and SPCs. Compared with synchronous patients, metachronous patients had an earlier pathologic stage of GC and SPCs. Our finding was consistent with those of *Watanabe et al. (2012)*, who reported that metachronous GC, together with colorectal cancer, tended to occur more often in patients with early stage of tumors. In addition, survival analysis showed that the 1-, 2-, and 3-year survival rates of DPCFGC patients were 61.54%, 48.72%, and 28.21%, respectively. A Korean epidemiological study reported that the overall 5-year relative survival of GC patients was 55.7%–77% over the past 10 years (*Park et al., 2022*). By contrast, our study demonstrated a lower survival rate. A survival analysis of MPMNs patients with GC as SPC in northern Portugal revealed that the 10-year cumulative mortality rate of MPMNs patients was at least 1.5 times higher than that of patients with single FPC and was not associated with the location of FPC (*Morais et al., 2020a*). Thus, MPMN patients may have a worse survival status than single cancer patients. *Wu et al. (2006)* speculated that this result may be attributed to the occurrence of interstitial reactions in MPMN patients.

Cox regression analysis showed that the pathological stage of GC, whether to operate for GC, diagnostic interval, and pathological stage of SPCs were significant risk factors that influenced the survival time of DPCFGC patients. Kaplan–Meier analysis showed that the more advanced the pathological stage of GC, the non-radical surgery for GC, the diagnostic interval being less than 6 months, and the more advanced the pathological stage of SPCs, the shorter the OS of DPCFGC patients. These results are similar to those of previous studies. *Hanisch & Batsis (2011)* demonstrated that the advanced stage of GC and lymphatic vascular invasion of colorectal cancer were independent risk factors affecting OS and progression-free survival in GC patients with colorectal cancer. *Ha et al. (2007)* revealed that the 5-year survival rates of stage I, II, and III GC patients with SPCs were 61%, 39%, and 30%, respectively. In our study, the survival status of metachronous patients was significantly better than that of synchronous patients. *Eom et al. (2008)* concluded that the prognosis of metachronous patients was better than that of synchronous patients, and the main reason for this finding was that concurrent-onset cancers might have a significant negative impact on the overall medical status of patients, thus hindering FPC treatment to a certain extent. By contrast, heterochronic-onset cancers did not change the management of FPC per se and had a relatively smaller impact on patients' survival. Our study finding corroborated this view to some extent. Compared with synchronous patients, metachronous patients had a higher radical resection rate for both GC and

SPCs. Additionally, poor lifestyle may also influence the prognosis of MPMN patients. *Morais et al. (2019)* reported a relative mortality HR of 1.73 for MPMN patients who always consumed alcohol *versus* never consumed alcohol and that of 1.31 for overweight *versus* normal-weight MPMN patients. *Chen et al. (2015)* found that smoking increased the risk of death from tobacco-related cancers. However, smoking, alcohol consumption, and body weight showed no significant impact on the prognosis of DPCFGC patients in our study, which may have been due to the small sample size.

The nomogram is a visualization tool for clinical prediction and is widely used in the medical field owing to its advantage in achieving individualized clinical evaluation (*Bianco Jr, 2006*; *Guillonneau, 2007*). In this study, we established a nomogram for predicting the prognosis of DPCFGC patients based on significant variables in multivariate Cox regression analysis. According to the different classifications of variables, points were projected upward to obtain the score of variables. All scores were summed to obtain total scores, and then the points were projected downward, corresponding to the survival rate of patients. The higher the scores, the worse the survival prognosis (*Iasonos et al., 2008*). The nomogram developed in our study could predict the 1-, 2-, and 3-year survival rates for different patients according to their own clinical conditions, which facilitated the individualized assessment of patients by clinicians. Time-dependent ROC curves revealed that the efficacy in predicting 1-, 2-, and 3-year OS in DPCFGC patients was 0.922, 0.935, and 0.796, respectively, suggesting that the prognostic model was more effective in predicting early OS in DPCFGC patients. Calibration curves and DCA of the model showed good consistency and clinical applicability, suggesting that the nomogram was reliable. Based on the optimal cutoff value for risk scores, we divided DPCFGC patients into high-risk and low-risk groups. Kaplan–Meier analysis showed that the OS of the high-risk group was significantly lower than that of the low-risk group, suggesting that this prognostic model could be used for early stratification of DPCFGC patients with poor survival and that this model will aid clinicians in formulating risk-matched clinical treatment options.

Our study had some limitations. To control for bias, our study included only DPCFGC patients and excluded patients with third-order and higher order primary cancers, which resulted in a small sample size. Additionally, limited by sample size, this study performed only internal validation of the established nomogram but did not perform external validation, which may have led to some bias in the results. Therefore, the results of this study should be verified in a multicenter prospective study with a larger sample size.

## CONCLUSIONS

During follow-up, clinicians should attach great importance to the screening of GC survivors, especially at early stage in older men within 1 year after diagnosis, and be alert to the possibility of occurrent digestive system malignancies. The nomogram constructed in this study will improve the clinical predictive ability for the prognosis of DPCFGC patients and provide a theoretical basis for the early clinical development of individualized treatment plans.

### Funding

This work was supported by the Postgraduate Innovation Fund of Gansu University of Chinese Medicine (No. 2021CX61). The funders had no role in study design, data collection and analysis, decision to publish, or preparation of the manuscript.

### Grant Disclosures

The following grant information was disclosed by the authors:
Postgraduate Innovation Fund of Gansu University of Chinese Medicine: No. 2021CX61.

### Competing Interests

The authors declare there are no competing interests.

### Author Contributions

- Bing Wang performed the experiments, analyzed the data, prepared figures and/or tables, authored or reviewed drafts of the article, and approved the final draft.
- Lu Liu conceived and designed the experiments, authored or reviewed drafts of the article, and approved the final draft.

### Human Ethics

The following information was supplied relating to ethical approvals (i.e., approving body and any reference numbers):

The medical ethics committee of Gansu Provincial Hospital approval to carry out the study within its facilities.

### Data Availability

The raw measurements are available in the Supplemental Files.

### Supplemental Information

Supplemental information for this article can be found online at http://dx.doi.org/10.7717/peerj.15278#supplemental-information.

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
