# Peer review of "Clinical characteristics and prognostic nomogram analysis of patients with dual primary cancers with first gastric cancer: a retrospective study in China"

_PeerJ, doi:10.7717/peerj.15278_

## Round 0.1 · original submission · Major Revisions

I have completed my evaluation of your manuscript. The reviewers recommend reconsideration of your manuscript following major revision. I invite you to resubmit your manuscript after addressing the comments below. When revising your manuscript, please consider all issues mentioned in the reviewers' comments carefully: please outline every change made in response to their comments and provide suitable rebuttals for any comments not addressed. Please note that your revised submission may need to be re-reviewed.

·

Basic reporting

Clinical characteristics and prognostic nomogram analysis of dual primary cancers patients with first gastric cancer: A retrospective study in China

I found the study interesting and relevant for the prognosis of dual primary cancers in patients with first gastric cancer. The study is well-designed and meticulously executed. The nomogram analysis from the present study can be useful for the early clinical development of individualized treatment plans. Following are the specific comments to further strengthen the manuscript,

1. Are the male population more vulnerable to the disease consequences and reoccurrence, please explain?
2. Please improve the quality of Figures 4 and 5. Consider increasing the font size in these figures for better readability.

Experimental design

Well designed.

Validity of the findings

Upto the mark.

·

Basic reporting

This manuscript by Wang and coworkers titled ‘Clinical characteristics and prognostic nomogram analysis of dual primary cancers patients with first gastric cancer: A retrospective study in China’ reports a nomogram for predicting the prognosis of double primary cancers with first gastric cancer (DPCFGC). This study is very exciting, and I commend authors for gleaning through the large cohort of samples to identify 78 DPCFGC patients with adequate clinical data information. The study is well designed, and introduction is written well and succinct. Results were described in detail although there is some scope for improvement. Authors discussed these findings in great details.

I ask authors to address the following comments.

1. Explain what variables were selected for nomogram and how were they selected. Please provide coefficients of each variable in regression formula or please write the formula.
2. Dichotomize the samples based on median risk score obtained from regression analysis performed for plotting nomogram and show that high risk scores for 1-, 2- and 3-year survival is associated with poor survival using survival curves plot.
3. Lines 166-173: Please show this data in a table or show graphical representation.
4. Line 95: What is ‘relevant data’? Please specify and elaborate. Please provide details of how many samples were excluded for lack of each of those data. This is important for the readers to understand if there is any bias in elimination that would invalidate the findings shown in this manuscript.
5. Please acknowledge that subgroup analysis may not have identified weak associations as the samples size is small.
6. Please acknowledge that it was a retrospective study and therefore, this nomogram needs to be validated in prospective studies.
7. I suggest authors to avoid writing ‘middle grade’. Did you intend to write ‘moderately differentiated’ instead?

Experimental design

No comment

Validity of the findings

No comment

Reviewer 3 ·

Basic reporting

The manuscript has some minor grammatical errors. I request the authors to correct them.

Experimental design

no comment

Validity of the findings

It is very difficult to understand the findings presented in figure 3 of this manuscript. I request the authors to present the results shown in figure 3 in an alternate format.

---

## Round 0.2 · accepted · Accept

It is a pleasure to accept your manuscript entitled " Clinical characteristics and prognostic nomogram analysis of patients with dual primary cancers with first gastric cancer: A retrospective study in China" in its current form for publication in PeerJ.

·

Basic reporting

This is upto mark.

Experimental design

Sound experimental design.

Validity of the findings

Finding of the study is valuable.

·

Basic reporting

Authors addressed my concerns. I have no further comments.

Experimental design

-

Validity of the findings

-

Reviewer 3 ·

Basic reporting

no comment

Experimental design

no comment

Validity of the findings

no comment

Additional comments

All comments have been addressed by the authors.